

# The 'what' and 'how' of screening for social needs in healthcare settings: a scoping review

Emma L. Karran[1], Aidan G. Cashin[2,3], Trevor Barker[1], Mark A. Boyd[4,5], Alessandro Chiarotto[6], Omar Dewidar[7], Jennifer Petkovic[7], Saurab Sharma[2,3], Peter Tugwell[8], G. Lorimer Moseley[1] and Identifying Social Factors that Stratify Health Opportunities and Outcomes (ISSHOOs) Collaborative Core Research Group,

[1] IIMPACT in Health, University of South Australia, Adelaide, South Australia, Australia
[2] Centre for Pain IMPACT, Neuroscience Research Australia, Sydney, New South Wales, Australia
[3] School of Health and Medical Sciences, University of New South Wales, Sydney, New South Wales, Australia
[4] Faculty of Health and Medical Sciences, University of Adelaide, Adelaide, South Australia, Australia
[5] Northern Adelaide Local Health Network, Adelaide, South Australia, Australia
[6] Department of General Practice, Erasmus University/Rotterdam, Rotterdam, Netherlands
[7] Bruyere Research Institute, University of Ottawa, Ottawa, Canada
[8] Department of Medicine and School of Epidemiology, University of Ottawa, Ottawa, Canada

Corresponding author
Emma L. Karran,
emma.karran@unisa.edu.au

## ABSTRACT

**Background:** Adverse social determinants of health give rise to individual-level social needs that have the potential to negatively impact health. Screening patients to identify unmet social needs is becoming more widespread. A review of the content of currently available screening tools is warranted. The aim of this scoping review was to determine *what* social needs categories are included in published Social Needs Screening Tools that have been developed for use in primary care settings, and *how* these social needs are screened.

**Methods:** We pre-registered the study on the Open Science Framework (https://osf.io/dqan2/). We searched MEDLINE and Embase from 01/01/2010 to 3/05/2022 to identify eligible studies reporting tools designed for use in primary healthcare settings. Two reviewers independently screened studies, a single reviewer extracted data. We summarised the characteristics of included studies descriptively and calculated the number of studies that collected data relevant to specific social needs categories. We identified sub-categories to classify the types of questions relevant to each of the main categories.

**Results:** We identified 420 unique citations, and 27 were included. Nine additional studies were retrieved by searching for tools that were used or referred to in excluded studies. Questions relating to food insecurity and the physical environment in which a person lives were the most frequently included items (92–94% of tools), followed by questions relating to economic stability and aspects of social and community context (81%). Seventy-five percent of the screening tools included items that evaluated five or more social needs categories (mean 6.5; standard deviation 1.75). One study reported that the tool had been 'validated'; 16 reported 'partial' validation; 12 reported that the tool was 'not validated' and seven studies did not report validation processes or outcomes.

# INTRODUCTION

It is well recognised that the broader circumstances of peoples' lives—the non-medical, contextual factors collectively referred to as the social determinants of health (SDoH)—play a major role in determining their health status. The SDoH include factors such as income, education, employment, housing, neighbourhoods, race, gender, working life conditions, and social connections; and they are shaped by the distribution of money, power and resources (*World Health Organisation, 2023*). These factors create a 'social gradient' that influences health outcomes. For example, the lower the socio-economic position of individuals or the communities in which they live, the worse their health (*Wilkinson & Marmot, 2003*).

Adverse SDoH give rise to individual-level 'social risk' factors that have the potential to negatively impact health. Social risks include factors such as housing instability, food insecurity, transportation barriers and social isolation; these factors are described as 'social needs' if they are prioritised by the individual to be of current concern (*Alderwick & Gottlieb, 2019*). While the SDoH are structural and very difficult to change, interventions to address social needs have been demonstrated to improve access to community resources, reduce healthcare spending and have the potential to confer health benefits (*Taylor et al., 2016*; *Yan et al., 2022*). Initiatives may involve tailoring or targeting healthcare; supporting care coordination; or initiating linkages to community-based support programs or social services that can provide needed resources (*Gottlieb et al., 2019*). Interest in integrating health care with social care in order to improve health outcomes is expanding—particularly in the United States (*Frier et al., 2020*; *Yan et al., 2022*) but also globally. The implementation of various conceptualisations of 'social prescribing' programs have recently been identified in 17 countries (*Morse et al., 2022*). Social prescribing recognises that factors such as finances, social relationships, and community engagement importantly impact health and well-being. 'Social prescription' involves referring patients to community-based activities and support services—based on identified social needs—and is often undertaken by 'link workers' in primary healthcare settings (*Drinkwater, Wildman & Moffatt, 2019*). Such 'non-medical' interventions are becoming increasing popular, reflecting factors such as an increased understanding of the wider determinants of health, the need to optimise health service use, decrease costs and reduce health inequalities (*Morse et al., 2022*; *Rempel et al., 2017*).

Identifying unmet social needs in clinical settings is commonly aided by the application of social needs screening tools (SNSTs). A systematic review of screening tools evaluating social risk factors in clinical settings developed in the United States of America identified 21 unique tools published prior to May 2018 (*Henrikson et al., 2019*). Undertaking appropriate development and validation procedures is 'gold standard' when incorporating screening tools into clinical care (*Poirier et al., 2022*); however, this review found that the

psychometric and pragmatic evaluation of the included tools was scarce. Consequently, the authors were unable to recommend any risk screening tools for widespread dissemination (*Henrikson et al., 2019*).

The expanding global interest in identifying and addressing social needs as a component of clinical care (*Morse et al., 2022*) and the suggestion that the conduct and publication of appropriate validation procedures is suboptimal (*Henrikson et al., 2019*), suggests that a review of the content of currently available screening tools—without geographic limitation, is required. The primary aim of this scoping review was to determine *what* social needs categories are included in published SNSTs that have been developed for use in primary care settings and *how* these social needs are screened.

## METHODS

We developed the protocol for this review in line with best-practice guidance for the development of scoping reviews (*Peters et al., 2022*) and registered our protocol *a priori* on the Open Science Framework (https://osf.io/dqan2/). The registered protocol outlines a three-stage scoping review that aims to develop a comprehensive set of items used to identify SDoH in clinical and research settings. This manuscript reports stage one of the broader project. We have reported this review in accordance with the PRISMA extension for scoping reviews (*Tricco et al., 2018*).

### Search strategy

We systematically searched MEDLINE and Embase from 01/01/2010 to 3/05/2022 to identify eligible studies. This date limitation avoided the identification of studies undertaken prior to the development of contemporary awareness of health equity— informed by the 2008 PROGRESS Plus Framework (*Kavanagh, Oliver & Lorenc, 2008*), and the 2008 report of the WHO Commission on SDoH (*Marmot et al., 2008*). The search strategy (Appendix 1) was composed of terms related to SDoH (including social determinants, social risks, social needs, social factors) and screening tools (including screen, identify, evaluate, assess). English search terms were used to identify studies, however there were no language restrictions to eligibility. In addition to this search strategy (outlined *a priori*), we conducted a subsequent search stage following completion of the screening process, that was not described in the registered protocol. We searched grey literature by conducting additional (focused) searches in Google and Google Scholar. We listed the names of SNST that were used or referred to in excluded studies (including reviews) and searched for these screening tool names individually. We evaluated any additional published studies that used these tools for their eligibility for inclusion in this review, according to the selection and screening process outlined below.

### Selection criteria

To be included in this review, the 'multi-domain' SNST tool "needed to be designed for use in primary healthcare settings to ensure its relevance for informing clinical care or social intervention as opposed to other purposes (*e.g.*, research) (*Henrikson et al., 2019*)." We defined a "primary healthcare setting" as a clinical setting in which a person would usually

have their first encounter with the health system. The tool needed to include at least 1 social risk screening question in two or more of the following domains: economic stability (*e.g.*, employment, income, and expenses), education (*e.g.*, early childhood education, education level, and literacy), social and community context (*e.g.*, social support systems, community engagement, and immigrant/refugee status), health and clinical care (*e.g.*, access, coverage, and provider availability and cultural competence), neighbourhood and physical environment (*e.g.*, housing, transportation, and safety and crime), and food (*e.g.*, food insecurity and access to healthy options).

We included studies that assessed a SNST in any population regardless of their age, language or other socio-demographic characteristics and reported this tool in adequate detail. Studies were excluded if they did not report the screening tool items in adequate detail—*i.e.*, both the question and response options needed to be provided in the published or publicly available material. Study protocols, reviews, editorials, commentaries and conference abstracts were excluded.

## Screening and data extraction

Articles identified in the database search were retrieved and exported into EndNote citation management software (EndNote 20 (2013); Clarivate Analytics, Philadelphia, PA, USA), and then imported into Covidence systematic review management system (*Veritas* Health Innovation Limited, sydney, NSW, Australia) where duplicates were removed. Two reviewers (ELK and AGC) independently screened the titles and abstracts of all articles for relevance according to the inclusion and exclusion criteria. We then obtained the full text of potentially eligible studies, and the articles were further screened in duplicate (ELK and AGC) for eligibility, with reasons for exclusion recorded. Any discrepancies or disagreements between the two reviewers were discussed and a third reviewer was consulted to resolve any conflicts (if required).

Two reviewers (ELK and AGC) extracted the relevant data from each study using a standardised and pilot-tested excel spreadsheet. Ten percent of the included studies were extracted in duplicate and cross-checked for accuracy. The data extraction template included the following fields: country, study population, clinical setting, screening tool name, validation details, and SNST items. Details of the SNST items (*i.e.*, the questions and response options) were inserted into the excel spreadsheet under the headings of education, economic stability, social and community context, health and clinical care, neighbourhood and physical environment, food, utility needs, transportation barriers, interpersonal violence or safety, and 'uncategorised.' These headings were consistent with the categorisation suggested by *Henrikson et al. (2019)*.

## Data synthesis

We summarised the characteristics of included studies descriptively and calculated the number of studies that collected data relevant to each of the social needs categories. We referred to the extracted study-level data relating to the development and validation processes to classify the screening tools as 'validated' (as reported by the study authors), 'partially validated' (some validation procedures reported), 'not validated' (as reported by

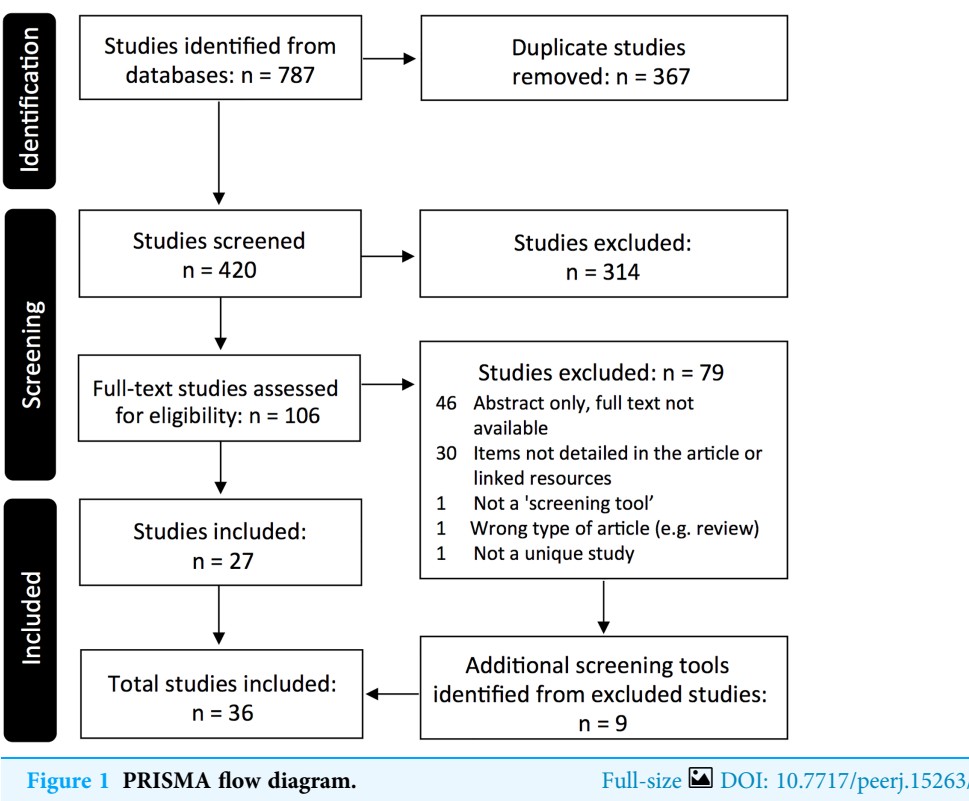

**Figure 1 PRISMA flow diagram.**

study authors) or 'validation details not reported.' To synthesise '*how*' the screening questions were asked we identified sub-categories to classify the types of questions relevant to each of the main social needs categories and tabulated the results, with examples.

## RESULTS

Our search identified 786 citations. After removing duplicates, we screened 420 unique citations, from which 106 full-text studies were assessed for eligibility. A total of seventy-nine studies were excluded (with reasons recorded); 27 studies were included (Fig. 1). Our Google and Google scholar search resulted in the inclusion of nine additional studies. This process has been described under 'search strategy' of the methods section.

Twenty-one (58%) of the included studies had been published since 2020; publication of the remaining 15 studies spanned 2012–2019. With the exception of one Australian study (*Andermann, 2018*; *Browne-Yung et al., 2019*), all of the included studies were undertaken in the United States of America. The key characteristics of these studies are summarised in Table 1. Forty-seven percent of studies were conducted in paediatric and child health settings (14 in non-emergency paediatric healthcare facilities and three in emergency departments), 25% were conducted in mixed, 'all-age' settings (*e.g.*, primary care clinics or general emergency departments), and 22% were conducted in adult healthcare settings. Of the studies conducted in mixed/adult settings, 15 were conducted in non-emergency healthcare facilities; and two were conducted in emergency departments. Two studies involved the development of a tool for clinical use but included non-patient populations.

**Table 1 Key characteristics of included studies.**

| Citation | Study population & setting | Screening tool name (if applicable) | Tool validation | Outline of tool development/validation |
|---|---|---|---|---|
| *Barcelos Winchester (2019)* | Delphi panel including experts in social work, nursing, public health, and psychology | Unnamed | Partially validated | Delphi process to develop, face validity established |
| *Bechtel et al. (2022)* | English speaking clinic patients aged ≥18 years attending a Federally Qualified Health Center in a large metropolitan city | **Core 5** screening tool | Partially validated | The public health nurses established content validity of the Core 5 screening checklist in collaboration with a working group of 17 nurse experts in the SDoH |
| *Beck, Klein & Kahn (2012)* | Electronic health record review of 639 infants seen within the first month of life for Well Child Care encounters at an urban hospital-based Paediatric Primary Care Center | **Social history template** | Not validated | Developed after a review of the literature and in consultation with medical staff, social services, and the medical–legal partnership at the Paediatric Primary Care Center |
| *Berkowitz et al. (2016)* | Primary care patients at two hospital-based primary care practices with existing complex care management programme for high-risk Medicare patients | **Health leads** | Validation not specified | No validation details provided |
| *Berkowitz et al. (2021)* | Data collected from a system-wide electronic health record platform from adult patients attending the Internal Medicine or Family Medicine departments of a large health service. | Unnamed ("SDoH pilot questionnaire") | Validation not specified | N/A |
| *Berry et al. (2020)* | 28 'key informants', including leadership personnel, frontline staff, volunteers & primary care providers from two adult outpatient clinics and one paediatric clinic | Unnamed ("SDoH screening & referral program") | Validation not specified | N/A |
| *Bittner et al. (2021)* | Patient data were extracted from the electronic health records of patients with a care provider at an independent practice affiliated with the Boston Children's Hospital | **HNA** screening tool | Partially validated | THE HNA includes previously (individually) validated questions (from the Health Leads screening toolkit) |
| *Bradywood, Watters & Blackmore (2021)* | Back pain patients seen in a neurosurgical clinic within an urban, tertiary-care hospital who progressed to surgery | **Core 5** screening tool | Partially validated | Questions written at a fifth-grade reading level. Pilot studies found consensus in usability, reported increases in social support referrals for patients and documented reliability in measurement. |
| *Browne-Yung et al. (2019)* | Focus group participants involving consumers recruited from an independent volunteer health consumer group (most were retired allied health professionals) | Unnamed ("Social health screening tool") | Not validated | The article outlines a three-phase development process involving focus groups and interviews |
| *Chagin et al. (2021)* | Patients receiving COVID-19 vaccines at multiple MetroHealth clinic locations. Screening occurred (1) in-person or by telephone, or (2) online through a patient portal questionnaire | Unnamed ("SDoH questionnaire") | Partially validated | The questions were obtained from previously validated surveys |
| *Ciccolo et al. (2020)* | English and Spanish speaking adult patients and parents of paediatric patients with adequate or limited health literacy attending a large, urban Emergency Department | Adapted from the **AHC-HRSN** screening tool | Not validated | The goal of this study was to develop and optimize a social risk and social need screening tool for Emergency Department patients |
| *Colvin et al. (2016)* | Paediatric interns at a 265-bed children's hospital. | **IHELLP** Questionnaire | Partially validated | This study 'sought to determine the validity of screening for social needs'. Found high positive predictive value of IHELLP, but lower negative predictive value. |
| *David et al. (2021)* | Retrospective chart review at paediatric clinic, involving families of children 1–17 years who presented for in-person well child visits. The clinic serves an ethnically diverse, predominantly low-income patient population. | Unnamed | Validation not specified | N/A (available in English & Spanish) |
| *Damas et al. (2022)* | Adult patients with a diagnosis of IBD seen at one of the three gastroenterology clinics. | **Social Barriers Score** | Not validated | N/A (no validation) |
| *de la Vega et al. (2019)* | All new primary care patients at an urban, tertiary care academic medical cente; approximately 50% of patients are insured by Medicaid. | **THRIVE** Screening Tool | Not validated | N/A (no validation) |
| *De Marchis et al. (2019)* | Patients and adult caregivers of paediatric patients attending primary care and Emergency Department settings that served a minimum of 30% publicly insured or uninsured patients | **AHC-HRSN** screening tool | Validation not specified | N/A |
| *Freibott et al. (2021)* | Eight staff members who conducted the screening using the CSHI assessment tool participated in in-depth interviews. Five staff (from four hospitals) completed an electronic survey | **CSHI** assessment tool | Not validated | N/A ("no validation process for the screening procedures") |

## Table 1 (continued)

| Citation | Study population & setting | Screening tool name (if applicable) | Tool validation | Outline of tool development/validation |
|---|---|---|---|---|
| *Friedman et al. (2021)* | Involved paediatric doctors, nurses, medical assistants, registration staff, social work, mental health providers & practice leadership. A total of 71% of the patient population in the neighbourhood is Hispanic, 48% are immigrants, >25% have household incomes below the poverty level. | Unnamed | Not validated | N/A Questions focused on SDoH were drawn from the American Academy of Paediatrics and The Joint Commission guidelines) |
| *Gottlieb et al. (2014)* | Adult caregivers seeking treatment of a child in a large, urban children's hospital Emergency Department. A total of 70% of patients have Medicaid insurance; 33% of patients are African American and 42% Latino. | **Iscreen** | Partially validated | Some items based on existing validated surveys |
| *Hensley et al. (2017)* | Children and families attending a primary care clinic; largely serving patients in households with income at or less than the 200% poverty level | **Health Begins** | Validation not specified | No validation details provided |
| *Kausar et al. (2022)* | All patients admitted to one of the 12 hospitals who had at least one documented social risk factor, except for patients admitted to maternity/paediatrics. | Unnamed ("Social risk factor screening module") | Partially validated | A committee of health care professionals and administrative personnel developed the screening module, with consideration of best practices, validated survey questions, and the communities served by the hospital. |
| *Kusnoor et al. (2018)* | Adult, English-speaking patients, presenting to an urban community health care clinic serving an underinsured population | **PRAPARE** | Validated | When possible, questions included in PRAPARE were obtained from validated instruments. A PRAPARE validation fact sheet has since been developed outlining 'gold standard' instrument validation. |
| *Macias-Konstantopoulos et al. (2022)* | Emergency Department patients (or parents/legal guardians of paediatric-age patients) of a large urban academic medical centre who were English- or Spanish-speaking | Unnamed | Validation not specified | The tool was developed for the primary care setting by the health system in which the study was conducted (available in English and Spanish). |
| *Mayo et al. (2022)* | Families of children receiving care at the paediatric outpatient clinic in the prior year. Children in the sample predominantly identified as Black or African American (90%) and 91% were insured with public insurance (Medicaid). | Adapted from **SEEK** | Partially validated | The study screening tool was adapted from a validated screening tool (the SEEK questionnaire) and was reviewed by an expert in the field |
| *Ovalle et al. (2021)* | Data obtained from the electronic health record at a Children's Hospital Medical Center, on children younger than 13 years attending for a well-child visit. | **SRS** Questionnaire (EHR-embedded) | Not validated | Developed through review of validated questions and screens (*i.e.*, the Survey on Income and Program Participation, the Children's Food Security Scale, Primary Care Evaluation of Mental Disorders Procedures, and the Partner Violence Screen) and local consensus of primary care physicians, nurses, social workers, and legal advocates |
| *Page-Reeves et al. (2016)* | 3,048 patients attending the family medicine clinics that serve a large, low-income population | **Well Rx** | Partially validated | The 11-item questionnaire (available in English or Spanish), was 'pre-tested' and conformed to low literacy needs |
| *Power-Hays et al. (2020)* | Universal screening for SDoH occurred in the paediatric hematology clinic at a large medical centre. The screening tool was offered in English, Spanish, and Haitian Creole, if low literacy—clinic staff read the screener to families. | **WECARE** | Validated | Previously validated in BMC's paediatric primary care network |
| *Ray et al. (2020)* | Caregivers of children ≤5 years presenting to a paediatric ED. | Unnamed | Partially validated | Items were synthesised from previously validated tools (WECARE and PRAPARE); also assessed social support with questions from the Medical Outcomes Study social support survey. |
| *Rinehart et al. (2021)* | The outpatient clinic serves a diverse paediatric population (age 0–21 years) that is 51% Hispanic, 40% African American, and 9% mixed race/ other race (internal data.) Most patients (90%) are insured by Medicaid. | Unnamed ("The screener") | Partially validated | Screening questions were developed with input from faculty and staff. Many were adapted from SDoH screening tools currently used in the USA (including HealthLeads and Hunger Vital Signs). Focus groups were conducted with providers and caretakers in the early stages of development, and screening questions were modified in response to feedback. |
| *Selvaraj et al. (2019)* | Participants were English- and Spanish-speaking parents or guardians of 2-week to 17-year-old children attending wellchild visits at paediatric primary care sites. | **ASK Tool** | Partially validated | The ASK Tool was mainly developed from validated questions in the literature and the WeCare tool. The child resilience question was not validated and led to a large number of false positives. |

(Continued)

| Citation | Study population & setting | Screening tool name (if applicable) | Tool validation | Outline of tool development/validation |
|---|---|---|---|---|
| *Sokol et al. (2021)* | Parents of paediatric patients (prior to a health maintenance paediatric examination). In some clinics, if the patient was aged ≥ 11 years, the patient completed the screening tool | Unnamed ("SDoH screening tool") | Partially validated | Questions to assess food insecurity were validated items (*Hager et al., 2010*). Questions to assess housing insecurity, utility insecurity, financial strain, transportation needs, employment needs, elder or child-care needs, and literacy needs were adapted from prior tools, including the PREPARE tool. |
| *Sundar (2018)* | Stakeholders in a primary care clinic in a suburban community. The population is racially diverse and about 17% live below the poverty line. | **YCLS** | Partially validated | Not validated for primary care |
| *Tedford et al. (2022)* | English- and Spanish-speaking caregivers of patients <18-years-old presenting to a children's hospital. | **p-SINCERE** | Not validated | Adapted the SINCERE tool for the paediatric population (p-SINCERE). The adjustments entailed including child-inclusive language, but no structural changes to the were made as the p-SINCERE tool was still administered to the adult caregivers. |
| *Tong et al. (2018)* | 17 primary care clinicians from 12 practices within one health system | Unnamed | Not validated | Adapted from National Academies social needs measures and supplemented with additional questions from the Hennepin County Life-style survey. |
| *Uwemedimo & May (2018)* | Caregivers accompanying patients aged 0–18 years attending a General Pediatric Practice for a preventive care visit. Patients are ethnically and socio-economically diverse; >2/3 receive health insurance coverage through Medicaid. | **FAMNEEDS** | Partially validated | Designed after review of published tools. Multiple iterations & formats were tested with caregivers to assess understanding of question constructs, frequency and time duration for completion, particularly among LEP and immigrant families. |
| *Wallace et al. (2020)* | Patients discharged from a large academic centre emergency department | Unnamed | Not validated | Development involved adapting 10 existing questions recommended by Health-Leads in their Social Needs Screening Toolkit. |

**Note:**

AHC, Accountable health communities health-related social needs; Core 5, SDoH screening checklist; CSHI, connecticut social health initiative assessment tool; HNA, health needs assessment screening tool; IHELLP, income, housing, education, legal status, literacy, personal safety questionnaire; PRAPARE, protocol for responding to and assessing patients' assests, risks, and experiences; p-SINCERE, screener for intensifying community referrals for health, paediatric version; SEEK, safe environment for every kid parent questionnaire; SRS, social risk screening questionnaire-EHR-embedded; WECARE, well child care, evaluation, community resources, advocacy, referral, education system; YCLS, your current life situation.

We classified the questions included in each screening tool in accordance with *Henrikson et al. (2019)* and summarised the frequency with which items corresponding to these categories were included in the SNSTs in Fig. 2. Questions relating to food insecurity and the physical environment in which a person lives were the most frequently included items (92–94% of tools) followed by questions relating to economic stability and aspects of social and community context (81% of tools). Seventy five percent of the screening tools included items that evaluated five or more social needs categories (mean number of categories 6.5; standard deviation 1.75). An overview of the types of questions included under each category is provided in Table 2. In addition, many questionnaires asked respondents directly about immediate social needs—either in direct relationship to a social needs category: *e.g.*, "Do you need help finding a job?" or through a general question at the completion of the survey: *e.g.*, "Would you like assistance with any of the above?".

Only one study reported that the screening tool had been 'validated' (*Power-Hays et al., 2020*). Sixteen studies reported some validation procedures (*e.g.*, face or content validity, or the inclusion of items that had been previously validated)—we classified these screening tools as 'partially validated'. Twelve studies reported that the screening tool was 'not validated' and seven studies did not report any validation processes or outcomes.

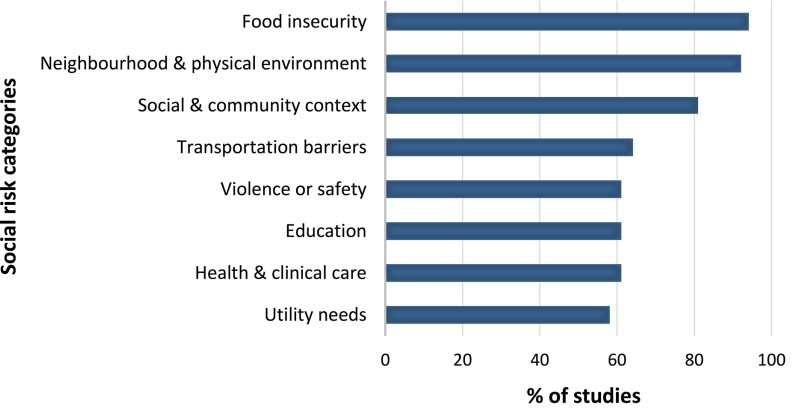

**Figure 2 Type of social risk categories evaluated and frequency of inclusion.**

# DISCUSSION

This scoping review provides an overview of social needs screening tools developed for use in clinical settings. The majority of tools included screening questions that addressed five to eight categories of social risks; over 80% of tools screened food insecurity, neighbourhood and physical environment, economic stability, and aspects of social and community context.

This review supports the increasing interest in screening for social needs in clinical settings. More than half of the studies included in this review have been published within the past 3 years; and 72% have been published since a previous systematic review of social risk screening tools (published in 2019) (*Henrikson et al., 2019*). It is relevant to note that this prior review searched electronic databases for US studies only. In contrast, we did not limit our searches by geography or language, however we identified only one eligible study undertaken outside of the United States (*Browne-Yung et al., 2019*). While it appears that implementing SNSTs in clinical settings is predominantly of interest in American settings, our review did exclude some studies of SNSTs conducted in Korea and Switzerland (that did not meet our eligibility criteria); and a qualitative validation of a SDoH screening tool in an Australian hospital setting (*Poirier et al., 2022*) has recently been published—suggesting worldwide interest.

While evaluating the reporting the content validity of the included tools was outside the scope of this study, our review reiterates concerns (*Henrikson et al., 2019*) that there is an important gap in the field when it comes to validating the instruments that are used to identify social needs in clinical settings (*Henrikson et al., 2019*). Only one included study reported using a 'previously validated' tool: the WECARE (Well child care, Evaluation, Community resources, Advocacy, Referral, Education) instrument (*Power-Hays et al., 2020*), however, we were unable to find specific details of the instrument validation procedures and results. We found a 'PRAPARE validation fact sheet' (available online) outlining 'gold standard' validation of the PRAPARE (Protocol for Responding to and Assessing Patients' Assets, Risks, and Experiences) instrument used by *Kusnoor et al. (2018)*; however, it appears that full validation details are yet to be published. Publication of

**Table 2 Question types and their frequency of inclusion, with examples.**

| Category (No. of items) | Example question | Response options |
|---|---|---|
| **Education** | | |
| Educational attainment (4) | What is the highest level of education you completed? | ☐ No formal schooling ☐ Primary school ☐ Secondary school (high, tech *etc.*) ☐ Tertiary ☐ Other (please specify): ___ |
| Literacy and health literacy (10) | Do you ever need help understanding what your doctor tells you, or help reading health information? | ☐ Yes ☐ No |
| Child education (11) | Do you have concerns about your child's learning or school performance? | ☐ Yes ☐ No |
| **Economic stability** | | |
| Employment status (14) | What is your current work situation? | ☐ Unemployed ☐ Part-time or temporary work ☐ Full-time work ☐ Otherwise unemployed but not seeking work (ex: student, retired, disabled, unpaid primary care giver) ☐ Other, please write: _______ ☐ I choose not to answer |
| Financial stability (27) | How would you say you are managing financially at the moment? How hard is it for you to pay for the very basics like food, housing, medical care, and heating? | ☐ Living comfortably ☐ Getting by ☐ Finding it difficult ☐ Not hard at all ☐ Not very hard ☐ Somewhat hard ☐ Hard ☐ Very hard ☐ Prefer Not to Answer |
| Income (1) | What is your average weekly household income? | __ (select option *e.g.* ☐ $25,000–$39,999 ☐ $40,000–$55,000….) |
| Financial support (2) | What is your primary source of financial support? | ☐ Self ☐ Spouse/Partner/Significant Other ☐ Parent ☐ Disability (SSI or SSD) ☐ Government/Public Assistance ☐ Social Security Retirement ☐ Retirement (not social security) ☐ Other (please specify) ☐ Do not know/unable to answer |
| **Social and community context** | | |
| Marital/living status (7) | Which of the following best describes your living arrangements? | ☐ Live with partner only ☐ Live with partner and children ☐ Sole parent with children ☐ Live with parents/other related adults ☐ Live with unrelated adults ☐ Other, please specify: _ |
| Social support—personal (24) | In a typical week, how many times do you talk on the telephone with family, friends, or neighbours? | ☐ Never ☐ Once a week ☐ Twice a week ☐ 3 times a week ☐ More than 3 times a week |
| Social support—community (24) | How often do you attend meetings of the clubs or organizations you belong to? | ☐ Never ☐ 1 to 4 times per year ☐ more than 4 times per year |
| Community programs (10) | In the last 12 months, have you received assistance from an organization or program to help you with any of the following? | ☐ None ☐ At least 1: ☐ Transportation ☐ Paying utility bills ☐ Education ☐ Food ☐ Paying for medications ☐ Childcare ☐ Housing ☐ Job search or training ☐ Care for elder or disabled |
| **Health and clinical care** | | |
| Access and affordability (19) | Have you needed to see a doctor but could not because it costs too much? | ☐ Yes ☐ No |
| Health insurance (4) | How stressful do you find concerns about lack of health insurance or inadequate insurance? | ☐ Not at all stressful ☐ A little bit stressful ☐ Moderately stressful ☐ Very stressful ☐ Extremely stressful ☐ Issue listed is not relevant to my family |
| **Neighbourhood and physical environment** | | |
| Housing status (13) | Which of the following best describes your current living situation? | ☐ House ☐ Flat, unit or apartment ☐ Retirement village ☐ Caravan or mobile home ☐ Housing trust ☐ Supported accommodation ☐ Other, please specify:__ |
| Housing insecurity (35) | How concerned are you that you will not have a place to live sometime in the next 6 months? | ☐ Very concerned ☐ Somewhat concerned ☐ Not concerned ☐ I choose not to answer |

| Category (No. of items) | Example question | Response options |
| --- | --- | --- |
| Home environment/ safety (23) | Think about the place where you live. Do you have problems with any of the following? | Pests (mice or roaches), mould, no/not working smoke detectors, water leaks, no window guards? □ Yes □ No |
| Neighbourhood safety (3) | Do you feel safe in your neighbourhood? | □ Never □ Rarely □ Sometimes □ Often □ Always □ Not Applicable |
| **Food insecurity** | | |
| Food (in)sufficiency (50) | In the past month, did anyone in your family go hungry because there was not enough money? | □ Yes □ No |
| Healthy food access/ affordability (30) | How often is client able to afford to buy healthy food for themselves and family? | □ Never □ Rarely □ Sometimes □ Often □ Always □ Not Applicable |
| **Utility needs** | | |
| Difficulty paying bills (17) | In the past 12 months, have you worried that any utility (electric, gas, water or oil) would be shut off for not paying your bills? | □ Yes □ No □ Already shut off □ I prefer not to answer |
| Problems with utilities (5) | Are you currently having issues at home with your utilities, such as your heat, electric, natural gas or water? | □ Yes □ No |
| **Transportation barriers** | | |
| Impact on medical appointments (21) | In the last 12 months, has lack of transportation kept you from medical appointments or from getting medications? | □ Yes □ No |
| Other access barriers (8) | In the past 12 months, has lack of transportation kept you from meetings, work, or getting things needed for daily living? | □ Yes □ No □ Prefer not to answer |
| **Interpersonal violence or safety** | | |
| Safety—general (2) | Do you feel unsafe in your daily life? | □ Yes □ No |
| Safety at home (5) | Are you or your family worried about feeling safe in your home? | □ Yes □ No |
| Safety in relationships (30) | Do you feel that you and/or your children are safe in your relationships? | □ Yes □ No |
| Safety in neighbourhood (2) | Do you have concerns about safety in your neighbourhood? | □ Yes □ No |
| **Stress** | | |
| General stress (8) | Stress means a situation in which a person feels tense, restless, nervous or anxious or is unable to sleep at night because his/ her mind is troubled all the time. Do you feel this kind of stress these days? | □ All of the time □ Most of the time □ Some of the time □ A little of the time □ None of the time |

the results of validity testing can influence the extent to which screening tools are adopted in clinical settings, provide important understanding of limitations, and importantly contribute to improvements in screening approaches. That the vast majority of studies reported incomplete tool validation (at best) and that results of validation procedures undertaken for widely available tools are not accessible has two main implications—firstly that caution is warranted if implementing these tools to guide care practices; and secondly, that appropriate validation of SNSTs should be prioritised.

## STRENGTHS AND LIMITATIONS

This study was conducted and reported in accordance with best-practice recommendations for scoping reviews (Peters et al., 2022; Tricco et al., 2018). The protocol was pre-registered on the Open Science Framework with deviations transparently reported in the methods section of this article. The results of the searches were screened in duplicate (by ELK and AGC) to identify eligible studies. Data extraction was shared between the two reviewers; with 10% of the data extracted by each reviewer also undertaken in duplicate to check for accuracy and consistency. We acknowledge that not extracting all data in duplicate brings risk of error; however, in light of the aim of our review and the high consistency of our findings, this is unlikely to have impacted our findings. It is possible that our initial database searchers missed eligible studies, but we minimised this risk by conducting individual searches for screening tools that were reported in reviews and studies that did not meet our eligibility criteria. Additionally, a thorough assessment of the content validity of the screening tools identified was beyond the scope of this review. This limits the detail with which we are able to describe and report instrument validation, and the conclusions that can be drawn.

### Significance and future directions

Our synthesis of the 'what' and 'how' of social needs screening has the potential to guide ongoing research in this expanding field. Our results reveal much consistency in the categories of social needs screened by currently available tools (i.e., 'what' to screen), along with widespread variability in 'how' to screen. Several social needs categories were included in more than 80% of tools. While some categories of social needs appear to be under-represented in the included tools, for example safety concerns; and unmet education, healthcare or utility needs—social risk categories are known to inter-relate (Patel et al., 2020) and a clear understanding of their unique and relative importance as components of SNSTs is yet to be understood. Our summary of the range of factors assessed and the types of questions asked to explore these factors (Table 2) may be highly useful for researchers, clinicians or healthcare administrators aiming to develop SNST or incorporate social needs questions into patient screening protocols.

This review supports the important findings of a previous review (Henrikson et al., 2019) that few screening approaches have been adequately validated and/or reported, and that little progress has been made towards addressing this concern. Social needs screening tools with demonstrated reliability and validity are an integral component of strategies designed to identify and address individual social needs in clinical settings. Additionally,

screening tools must (ideally) be validated in the specific population of interest and in the setting in which they are implemented (*Ali, Ryan & De Silva, 2016*). The predominance of United States data in this review highlights that generalisability issues must be considered when implementing available tools in clinical settings in other countries. The wide variation in country-specific social challenges and the importance of cultural validation of screening tools suggests that there is much work to be done globally.

Despite the range of challenges identified, it appears clear that there is a level of widespread, evidence-supported momentum to continue to consider and develop care pathways that screen for (and address) the social determinants of health in clinical settings (*Andermann, 2018*; *de la Vega et al., 2019*; *Neadley et al., 2020*). Recent studies have demonstrated improvements in both social outcomes (*e.g.*, employment, housing stability, access to community resources) (*Garg et al., 2015*) and health outcomes (*e.g.*, reduced injury presentations and associated costs) (*Strong et al., 2016*) from such initiatives. In addition, *Andermann & CLEAR Collaboration (2016)* provide practical guidance for how clinicians can consider and address the SDoH in routine clinical practice—outlining the potential for positive impacts at the patient, clinical and community level (*Andermann & CLEAR Collaboration, 2016*). It is recognised, however, that this is an emerging area of clinical practice that requires much more research to understand how benefits can be maximised, risks minimised and health equity enhanced (*Davidson & McGinn, 2019*; *Silverstein, Hsu & Bell, 2019*).

A consensus-driven tool that can be readily adapted and validated for implementation across jurisdictions has the potential to enhance efforts to integrate social care into healthcare settings on a broad scale. Similarly, a uniform data collection tool providing guidance on what data *researchers* should collect relating to the SDoH, and how to collect it —has the potential to illuminate inequalities in health outcomes, facilitate data pooling and progress public health understanding. Such initiatives are likely to be key in progressing towards greater understanding of the potential to take action to address health inequities.

## CONCLUSIONS

This review explored the content of published social needs screening tools developed for use in clinical settings. We found much consistency in the categories of social needs screened by currently available tools (*i.e.*, 'what' to screen), and widespread variability in 'how' to screen. Our findings have the potential to highly useful for researchers, clinicians or healthcare administrators aiming to develop SNST or incorporate social needs questions into patient screening protocols. Global efforts to address health inequities through the integration of health and social care are likely to be enhanced by further efforts to develop valid and feasible tools.

### Funding

Emma L. Karran, Trevor Barker and G. Lorimer Moseley are supported by an Australian National Health and Medical Research Council Leadership Investigator Grant to G. Lorimer Moseley (NHMRC ID 1178444). Aidan G. Cashin is supported by an Australian National Health and Medical Research Council Emerging Leadership Investigator Grant (NHMRC ID 2010088). Saurab Sharma is supported by John J. Bonica Fellowship from the International Association for the Study of Pain. Peter Tugwell is funded by Canada Research Chair Program. The funders had no role in study design, data collection and analysis, decision to publish, or preparation of the manuscript.

### Grant Disclosures

The following grant information was disclosed by the authors:
Australian National Health and Medical Research Council Leadership Investigator: 1178444.
Australian National Health and Medical Research Council Emerging Leadership Investigator: 2010088.
International Association for the Study of Pain.
Canada Research Chair Program.

### Competing Interests

Emma L. Karran has received speaker fees for lectures on pain and rehabilitation from professional and scientific bodies, and reimbursement of travel costs related to presentations at scientific conferences/symposia. G. Lorimer Moseley has received support from: Reality Health, ConnectHealth UK, Institutes of Health California, AIA Australia, Workers' Compensation Boards and professional sporting organisations in Australia, Europe, South and North America. Professional and scientific bodies have reimbursed him for travel costs related to presentation of research on pain and pain education at scientific conferences/symposia. He has received speaker fees for lectures on pain, pain education and rehabilitation and conference travel support from Sequirus. He receives royalties for books on pain and pain education. G. Lorimer Moseley is an Academic Editor for PeerJ. Peter Tugwell has received consulting fees to provide independent medical consultation and professional services. He is an independent Committee Member for clinical trial Data Safety Monitoring Boards for FDA approved trials being conducted by UCB Biopharma GmbH & SPRL, Parexel International, Prahealth Sciences. Peter Tugwell is an [unpaid] Chair of the Management SubCommittee of the Executive Committee of a registered non-profit independent medical research organization, OMERACT. OMERACT receives unrestricted educational grants from the American College of Rheumatology, European League of Rheumatology and several pharmaceutical companies listed in this section, which is used to support fellows, international patient groups and support a major international bi-annual conference which results in many peer-reviewed publications. There are no competing interests for any other author.

## Author Contributions

- Emma L. Karran conceived and designed the experiments, performed the experiments, analyzed the data, prepared figures and/or tables, authored or reviewed drafts of the article, and approved the final draft.
- Aidan G. Cashin conceived and designed the experiments, performed the experiments, analyzed the data, authored or reviewed drafts of the article, and approved the final draft.
- Trevor Barker conceived and designed the experiments, authored or reviewed drafts of the article, and approved the final draft.
- Mark A. Boyd conceived and designed the experiments, authored or reviewed drafts of the article, and approved the final draft.
- Alessandro Chiarotto conceived and designed the experiments, analyzed the data, authored or reviewed drafts of the article, and approved the final draft.
- Omar Dewidar conceived and designed the experiments, analyzed the data, authored or reviewed drafts of the article, and approved the final draft.
- Jennifer Petkovic conceived and designed the experiments, analyzed the data, authored or reviewed drafts of the article, and approved the final draft.
- Saurab Sharma conceived and designed the experiments, analyzed the data, authored or reviewed drafts of the article, and approved the final draft.
- Peter Tugwell conceived and designed the experiments, authored or reviewed drafts of the article, and approved the final draft.
- G. Lorimer Moseley conceived and designed the experiments, authored or reviewed drafts of the article, and approved the final draft.

## Data Availability

The raw data extracted from the reviewed articles are available in the Supplemental File.

## Supplemental Information

Supplemental information for this article can be found online at http://dx.doi.org/10.7717/peerj.15263#supplemental-information.

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
