# Peer review of "The ‘what’ and ‘how’ of screening for social needs in healthcare settings: a scoping review"

_PeerJ, doi:10.7717/peerj.15263_

## Round 0.1 · original submission · Minor Revisions

We are grateful for the careful reading and productive feedback of the reviewers . Please revise the manuscript accordingly addressing in particular the search and stating the aim earlier.

Reviewer 1 ·

Basic reporting

The primary aim of this scoping review was to investigate what social needs categories are included in published that have been developed for use in the primary healthcare setting and how these social needs are screened for. While there seems increasing interest in screening for social needs in clinical settings, yet validity was rarely investigated.
The content of the article appears to be within the scope of the journal. The introduction section illustrates the context very well. Methods are described with sufficient detail and information. For example, a rationale is provided why articles after 2010 were considered. Sources were adequately cited and paragraphs were logically organized in coherent paragraphs and subsections. For reporting, the (PRISMA-ScR) Checklist was used which is adequate. As an emergency physician reviewer (and obviously working in the “primary healthcare setting” – see below) I am surprised how little EM – specific tools came up in your search, e.g. regarding violence and/or safety (I wonder why e.g. the Senior Abuse Identification (ED Senior AID) tool did not come up).

Experimental design

As an emergency physician reviewer (and obviously working in the “primary healthcare setting” – see below) I am surprised how little EM – specific tools came up in your search, e.g. regarding violence and/or safety (I wonder why e.g. the Senior Abuse Identification (ED Senior AID) tool did not come up, of note: I am not the author of this tools). Is it the search criteria that are very broad?

Or does EM not belong in the primary healthcare sector?
Setting: Please provide readers with a definition of “primary healthcare setting”; it might mean different things to different readers form different contexts. Please compare with PRISMA checklist item 4; population or participants, concepts, and context.

Validity of the findings

Setting: Please provide readers with a definition of “primary healthcare setting”; it might mean different things to different readers form different contexts. Please compare with PRISMA checklist item 4; population or participants, concepts, and context.

Additional comments

MINOR:
Please remove redundancies in lines 170 – 176: “Forty-seven percent of studies were conducted in paediatric and child health settings” and “Seventeen studies (47%) were conducted in paediatric settings”).
Very minor detail: Acronym in line 219: Is it PRAPARE or PREPARE?

Reviewer 2 ·

Basic reporting

o This is an important area in which greater understanding and progress are required. Therefore, the article provides a valuable contribution to increasing the body of evidence on this topic.
o The scoping review reads well with mostly good use of unambiguous and clear point making.
o Abstract
i. Clearly identifying the aim early in this review would strengthen the article. As it currently is, it is difficult to determine the exact aim of the review simply by reading the abstract. This would give the reader a quick understanding of the intention of the review. Perhaps the authors could add this to the background section or in the methods section of the abstract.
ii. Conclusion is sound and in line with the current global status of incorporating SDoH into clinical/individually based health care.
o Introduction
i. The authors have correctly identified the influence of suboptimal SDoH on health outcomes, and a need for SDoH screening at the clinical level.
ii. Line 74 – 76: this sentence appears a little randomly placed. It would be useful to add some context around ‘social prescribing’ i.e. what is it, who does it, brief examples. And also relate it to the argument of how integrating healthcare with social care to improve health outcomes is expanding globally.
iii. Line 83: add reference 8 to the end of this sentence.
iv. Lines 85- 86: it would be useful to add citations here that justify and support “that the conduct and publication of appropriate validation procedures is inadequate”. This is truly such a gap, just needs a little more citing to emphasise this point, and therefore better justifying the review.
v. Line 86: Delete “at least in the United States”. It is a worldwide problem, not just in the US. Also, perhaps instead of saying ‘inadequate’ the authors could use more positive language e.g. are suboptimal or requires progression/advancement/development/ (or something along those lines).
vi. Line 87: Perhaps instead of the word “timely” include something to suggest the current literature review would contribute to improving the “inadequacy” mentioned, it might be something as simple as changing it to; is required, or, is called for. The word timely just seems a little passive.

Experimental design

• Search strategy
o Lines 109 – 111: add a little more clarity to this sentence demonstrating how grey literature was also searched e.g. we searched grey literature by conducting Google and Google scholar searches.

• Selection criteria
o It would be useful to include a table displaying inclusion and exclusion criteria rather than describing them. This would probably also assist if there are any word limit restrictions.

Validity of the findings

• This is an important area of work. The findings support the global consensus that further research is needed and the body of knowledge around this topic increased.

• Results
Lines 162 – 165 is worded similarly to the methodology section, whereas it would be better to present this in a ‘results’ format. Perhaps say something like, our Google and Google scholar search resulted in the “inclusion of nine additional studies”. The process followed is already described under ‘search strategy’ of the methods section.

• Discussion
- Line 203: delete “at least in the United States” stating it too often becomes a bit repetitive. It is mentioned elsewhere throughout the paper so the point is still able to be made. If the authors decide to include the below references, this would demonstrate an even greater global interest in the screening of social needs in clinical settings. Especially as this is a point made by the authors on line 212.
-Line 206: reference.

o The importance of validation of SDoH screening tools is discussed. This may be an opportunity to elaborate more on reference 9, which is of relevance here. This may also help in addressing the below point.

o Overall the discussion could use a little more substance and broader inclusion of related literature. Perhaps if the authors review and include relevant information from the articles in the general comments section below, this may provide more depth to the discussion.

o Lines 214 – 215: 8 might be better placed at the end of the sentence rather than after the word concerns.

• Significance and future directions
Clear

• Conclusion
comprehensive and all inclusive of the review findings, yet to the point.

Additional comments

• This is an important area of work. Assessment of an individual social needs is an addition to clinical care that is necessary for health outcomes to improve.
• The article cites most of the major players in this field. However, there is a prominent Canadian researcher whose work in this area has not been cited in this paper.

Andermann A. Screening for social determinants of health in clinical care: moving from the margins to the mainstream. Public Health Reviews. 2018;39: Article 19. Available from: https://doi.org/10.1186/s40985-018-0094-7.

Andermann A. Taking action on the social determinants of health in clinical practice: a framework for health professionals. Canadian Medical Association Journal. 2016;188(17-18):E474-83. Available from: https://doi.org/10.1503%2Fcmaj.160177

Andermann A., on behalf of the CLEAR Collaboration. (2013). The CLEAR Toolkit: Helping Health Workers Tackle the Social Causes of Poor Health [version 3.0]. Retrieved from www.mcgill.ca/clear
Though her work may have been excluded from the studies in this scoping review, and her team have not developed an SDoH screening tool per se, it would be worthwhile incorporating her work in some way to increase the credibility of this paper. Particularly when suggesting there is worldwide interest and justifying the importance of incorporating SDoH into clinical care.

• In addition the below article is also related to reference 14.
Neadley K, McMichael G, Freeman T, Browne-Yung K, Baum F, Pretorius E, et al. Capturing the Social Determinants of Health at the Individual Level: A Pilot Study. Public Health Research & Practice. 2020.

• The below article also supports the argument of research dearth in this area, and also discusses the ‘what and how’.
Frier A, Devine S, Barnett F, Dunning T. Utilising Clinical Settings to Identify and Respond to the Social Determinants of Health of Individuals with Type 2 Diabetes-a Review of the Literature. Health & Social Care in the Community. 2019.

It is entirely up to the authors if they wish to include these citations depending if they deemed them relevant or not. It may add to the depth of the discussion (mentioned above).

• The tables (1&2) and figure 2 are clear and comprehensive and provide valuable and useful information.

---

## Round 0.2 · accepted · Accept

Thank you for your revision. Both reviewers are satisfied with the revisions and therefore we are happy to accept the paper.

Reviewer 1 ·

Basic reporting

no further comments - my previous comments were addressed

Experimental design

no further comments - my previous comments were addressed

Validity of the findings

no further comments - my previous comments were addressed

Additional comments

no further comments - my previous comments were addressed

Reviewer 2 ·

Basic reporting

Thank you all revisions are adequate.

Very minor comment/question: "becoming increasing popular" - Should this be 'increasingly' rather than 'increasing"

Author response: We agree with your suggestion. We have added the following:

Line 76: “Social prescribing recognises that factors such as finances, social relationships, and community engagement importantly impact health and well-being. ‘Social prescription’ involves referring patients to community-based activities and support services - based on identified social needs - and is often undertaken by ‘link workers’ in primary healthcare settings.8 Such ‘non-medical’ interventions are becoming increasing popular, reflecting factors such as an increased understanding of the wider determinants of health, the need to optimise health service use, decrease costs and reduce health inequalities.7, 9”

Experimental design

Thank you all revisions are adequate

Validity of the findings

Thank you all revisions are adequate

Additional comments

Thank you all revisions are adequate, and it is great to see this area gaining more and more traction.